# Assessment of the methodological quality of local clinical practice guidelines on the identification and management of gestational diabetes

Bridget Daley,[1] Graham Hitman,[2,3] Norman Fenton,[4] Scott McLachlan[4]

¹Centre for Genomics and Child Health, Queen Mary University of London, London, UK
²Barts Health NHS Trust, Diabetes and Metabolism, London, UK
³Barts and the London School of Medicine and Dentistry, Queen Mary University of London, London, UK
⁴EECS, Queen Mary University of London, London, UK

**Correspondence to**
Bridget Daley;
b.j.daley@qmul.ac.uk

## ABSTRACT

**Objective** Gestational diabetes is the most common metabolic disorder of pregnancy, and it is important that well-written clinical practice guidelines (CPGs) are used to optimise healthcare delivery and improve patient outcomes. The aim of the study was to assess the methodological quality of hospital-based CPGs on the identification and management of gestational diabetes.

**Design** We conducted an assessment of local clinical guidelines in English for gestational diabetes using the Appraisal of Guidelines for Research and Evaluation (AGREE II) to assess and validate methodological quality.

**Data sources and eligibility criteria** We sought a representative selection of local CPGs accessible by the internet. Criteria for inclusion were (1) identified as a guideline, (2) written in English, (3) produced by or for the hospital in a Western country, (4) included diagnostic criteria and recommendations concerning gestational diabetes, (5) grounded on evidence-based medicine and (6) accessible over the internet. No more than two CPGs were selected from any single country.

**Results** Of the 56 CPGs identified, 7 were evaluated in detail by five reviewers using the standard AGREE II instrument. Interrater variance was calculated, with strong agreement observed for those protocols considered by reviewers as the highest and lowest scoring based on the instrument. CPG results for each of the six AGREE II domains are presented categorically using a 5-point Likert scale. Only one CPG scored above average in five or more of the domains. Overall scores ranged from 91.6 (the strongest) to 50 (the weakest). Significant variation existed in the methodological quality of CPGs, even though they followed the guideline of an advising body. Specifically, appropriate identification of the evidence relied on to inform clinical decision making in CPGs was poor, as was evidence of user involvement in the development of the guideline, resource implications, documentation of competing interests of the guideline development group and evidence of external review.

**Conclusions** The limitations described are important considerations for updating current and new CPGs.

## INTRODUCTION

Diabetes, the most common metabolic disorder of pregnancy, carries multiple risks for both mother and baby.[1,2] Diabetes

### Strengths and limitations of this study

► Evaluation of each guideline by five reviewers.
► Comparative analysis of hospital-based guidelines.
► Use of an accepted, structured and validated tool.
► Inclusion of English-language only guidelines and exclusion of those published prior to 2013 mean those from non-English-speaking countries and some older well-known guidelines were not assessed.
► Use of a categorical rather than a statistical presentation of intra-domain results means that while results are more approachable, individual reviewer scores are not shown.

in pregnancy may be classified into pre-existing type 1 Diabetes, type 2 diabetes and gestational diabetes mellitus (GDM). GDM is defined by hyperglycaemia with onset or first recognition during pregnancy.[3,4] Between 40% and 60% of women diagnosed with GDM go on to develop type 2 diabetes in later life and are significantly more likely to require a caesarean delivery. Evidence suggests babies of diabetic mothers have an increased risk of childhood obesity and diabetes. The use of evidence-based guidelines for the prevention and treatment of GDM can reduce the adverse outcomes in pregnancy and childhood. The aim of the study was to assess the methodological quality of local clinical practice guidelines (CPGs) on the identification and management of GDM.

### Gestational diabetes mellitus

GDM occurs in 2%–25% of pregnancies[5,6] and, depending on the diagnostic criteria used, rates in the UK may be as high as 17%.[6,7] While the original definition for GDM was based on maternal risk for developing diabetes postpartum,[8] newer glucose criteria have been developed based on risk of maternal and neonatal complications.[8,9] While there have been five international workshops devoted to

GDM,[10] the diagnostic criteria for GDM remain an area of considerable debate.[11] The Hyperglycaemia and Adverse Pregnancy Outcomes (HAPO) study demonstrated a linear association between increasing levels of maternal hyperglycaemia and adverse perinatal outcomes, with no obvious threshold.[1] Diagnostic criteria based on the HAPO study were proposed by the International Association of Diabetes in Pregnancy Study Group (IADPSG) in 2010.[8] Their guidelines present diagnostic plasma glucose levels for before (fasting), and at 1 and 2 hours after administration of an oral 75 g glucose load that identify patients whose babies have nearly double the risk of three specific adverse clinical outcomes: macrosomia, increased adiposity and hypoglycaemia.[8] These diagnostic criteria have been subsequently adopted by the WHO, Canadian Diabetes Association (CDA)[12] and Australian Diabetes in Pregnancy Society (ADIPS).[13] However, they remain controversial and have not been supported by other bodies, and there has been no randomised control trial to test the efficacy of these criteria.[14] Furthermore, WHO have acknowledged they must be revisited in light of weak support for their diagnostic criteria, and newer studies on cost-effectiveness.[15] The same reasons were also cited by New Zealand's Best Practice Advocacy Centre (BPAC) when they persisted in using older, better supported diagnostic criteria.[16] In 2015 National Institute for Health and Care Excellence (NICE) published an updated guidance on diabetes in pregnancy that included recommendations on diagnostic thresholds for GDM that differ from those adopted by WHO.[14] Recent analysis suggests NICE thresholds are more cost effective than those of WHO, but this may be applicable only within the UK setting.[17] Table 1 summarises a comparison of screening and diagnostic criteria for diagnosing GDM as referenced or relied on by the CPGs in this study.

## Clinical practice guidelines

Reliance on the best evidence is fundamental to ensuring quality healthcare. Valid guidelines for clinical practice are a powerful tool and the key to informing evidence-based practice.[18 19] The Institute of Medicine defines CPGs as statements that include recommendations, intended to optimise patient care, informed by a systematic review of evidence and an assessment of the benefits and harms of alternative care options.[20] CPGs are underpinned by systematic review of evidence and are usually formulated by groups of stakeholders with relevant domain expertise. They may be developed at local, national and international levels, and International CPGs are often adopted nationally or locally, although with local clinical and demographic setting-specific alterations.[21] CPGs represent a convenient method for evaluating and assimilating evidence and presenting standardised recommendations to those tasked with delivering healthcare.[19]

Primary goals of CPGs are reductions in both variation in clinical practice and the expense of repeating inappropriate treatments.[22 23] Achieving this goal is said to improve the quality of care received by patients, while reducing the incidence of inappropriate treatment.[23 24] In 2008, the UK NICE systematically reviewed the evidence for effectiveness for various interventions to manage all types of diabetes during pregnancy including GDM. This review led to publication of a complete preconception to postnatal diabetes in pregnancy management guideline,[25] updated with more recent evidence and recommendations in 2015.[14] Others, such as New Zealand's BPAC, act as little more than localised extensions of NICE, even affording NICE oversight of all guideline adaptation efforts.[26] Such approaches may limit beneficial innovation of clinical practices, application to the local population and, when a particular guideline is found wanting, blind adherence may even increase potential harm and spread it over a significantly larger patient population.[27 28]

Since 2017, the authors have been involved in a research project (PAMBAYESIAN) to create a new generation of easy-to-use medical decision support systems for direct patient and clinician use.[29] One of the project's case

**Table 1** Comparison of screening and diagnostic criteria of gestational diabetes

| | Year | Patient screening | Two-step testing | Screening | Screening threshold | OGTT glucose load (g) | Dx thresholds (BGL in mmol/L) | | | Elevated OGTT values for Dx |
|---|---|---|---|---|---|---|---|---|---|---|
| | | | | | | | FBG | 1 hour | 2 hour | |
| NICE | 2015 | Clinical risk | | | | 75 | 5.6 | - | 7.8 | 1 |
| SIGN | 2017 | Clinical risk | | | | 75 | 5.1 | 10 | 8.5 | 1 |
| BPAC | 2014 | All | Y | 50 g GCT | 7.8 | 75 | 5.5 | - | 9.0 | 1 |
| CDA | 2013 | All | Y | 50 g GCT | 7.8 | 75 | 5.3 | 10.6 | 9.0 | 1 |
| IADPSG | 2010 | All | | | | 75 | 5.1 | 10.0 | 8.5 | 1 |
| ADIPS | 2014 | All, unless resources limited | | | | 75 | 5.1 | 10.0 | 8.5 | 1 |
| WHO | 2013 | All | | | | 75 | 5.1 | 10.0 | 8.5 | 1 |

ADIPS, Australian Diabetes in Pregnancy Society; BPAC, Best Practice Advocacy Group New Zealand; CDA, Canadian Diabetes Association; FBG, fasting blood glucose; GCT, glucose challenge test; IADPSG, International Association of the Diabetes and Pregnancy Study Groups; NICE, National Institute for Health and Care Excellence; OGTT, oral glucose tolerance test; SIGN, Scottish Intercollegiate Guidelines Network; WHO, World Health Organization.

studies focuses on GDM, and this Appraisal of Guidelines for Research and Evaluation (AGREE) II protocol review was undertaken to assess the quality of CPGs and help contribute to PAMBAYESIAN's decision support system.

While AGREE II has received some criticism[30] (and other tools for evaluating CPGs exist[31]), it is nevertheless an accepted and validated tool for assessing the methodological quality of CPGs[31]; it does not assess the implementation of the guideline. The AGREE II instrument[32] comprises 23 items arranged in six domains: (1) scope and purpose, (2) stakeholder involvement, (3) rigour of development, (4) clarity and presentation, (5) applicability and (6) editorial independence. Responses are scored on a Likert scale from 1 to 7 (1=strongly disagree, 7=strongly agree).

One of the motivations for AGREE II in general is to address the low rates of adherence to CPGs.[33 34] Studies have shown the number of health professionals using CPGs to be less than one-third.[33 34] Of those, around 60% failed to review patient records and followed the guideline so blindly that they repeated tests that had already been conducted, which increases health service wastage.[33 34] The AGREE II study's primary focus is on the form of the guideline: identification of who wrote the guideline, conflicts of interest for authors or from funding sources, the process for evaluation of evidence and so on. The AGREE II committee cite these as primary issues that affect the quality and reliability of CPGs and their effect on the care delivered in hospitals. The low usage rates observed coupled with increased resource wastage conflict with the CPGs primary purpose to such degree that issues of who wrote or funded its development, and the processes they relied on to select and evaluate the evidence may not actually be impacting on the quality of care to any significant degree.[33 34] No matter how well a guideline scores on the AGREE II protocol, AGREE studies are silent on how well the guideline is being applied in practice. Our evaluation accordingly does not take account of how well CPGs are applied by clinicians in the respective clinical environments.

## METHODS

We conducted a review of local CPGs addressing GDM with quality assessment using the AGREE II instrument.

### Guideline selection

Rather than seeking guidelines or guideline reviews from the literature, which in many cases only review the ideal guidelines of national or non-clinical organisations (such as those from medical associations, professional colleges, health insurance providers, government health departments and so on), we sought a representative selection of local CPGs developed or required to be used by clinicians in western hospitals where training and certification processes are similar to those where three of our authors trained and practiced. Given that hospitals generally do not publish local CPGs in the academic

literature, it was necessary to perform an internet search of government guideline clearinghouse and hospital public-access webservers using the search terms hospital, CPG, diabetes and pregnancy. The criteria for inclusion were that the document (1) was explicitly identified as a guideline, (2) was written in English, (3) was produced by or for the hospital in a western country, (4) included diagnostic criteria and recommendations concerning GDM, (5) demonstrated some evaluation or inclusion of evidence and (6) was easily accessible over the internet to health service consumers. In addition, we ensured (7) two CPGs from each country identified. Where more than two from any one country met the requirements, the ones having the most recent updates were used.

### Guideline quality assessment

To increase the reliability of the appraisal, the AGREE II protocol user manual recommends each guideline be assessed by at least two, but preferably four or more, appraisers.[32] Adhering to the AGREE II instrument protocol, each guideline in this study was independently reviewed by five assessors with a range of expertise (a neonatal paediatric nurse, a clinical midwifery research fellow, a diabetologist, a guideline and learning health systems researcher and a learning systems statistician, all with knowledge of evidence-based medicine). Reviewers did not communicate or confer about the guidelines during the review process. Reviewers were provided with the guidelines and any supporting documents or related publications. Responses were collected using a secure online survey tool that exported into an Excel spreadsheet prepared specifically to report per domain, per question and per reviewer. Percentages were calculated in Excel for each domain following the AGREE II protocol manual algorithm shown in figure 1. Additionally, reviewer variance scores were calculated across all scores for each question using Excel's built-in VAR function.

A precedent for categorical, rather than statistical, reporting of AGREE II scores has been set by Duda et al[35] and extended by Eady et al.[30] In order to make the scores more relevant to readers and enable fair comparison, our review reports the AGREE II outcomes using the 5-point Likert scale described in Eady et al[30] as excellent (>80%), good (>60%–80%), average (>40%–60%), fair (>20%–40%) and poor (≤20%).

### Patient and public involvement

Prior to commencing, the aims of the PAMBAYESIAN project including the diabetes component and the AGREE II protocol were presented to the North East London Diabetes Research Network Lay Panel who have helped to inform the research while commenting on the wide variation in patient experiences in services provided for

$$\frac{\text{Obtained score} - \text{Minimum possible score}}{\text{Maximum possible score} - \text{Minimum possible score}}$$

**Figure 1** AGREE II protocol domain scoring algorithm.

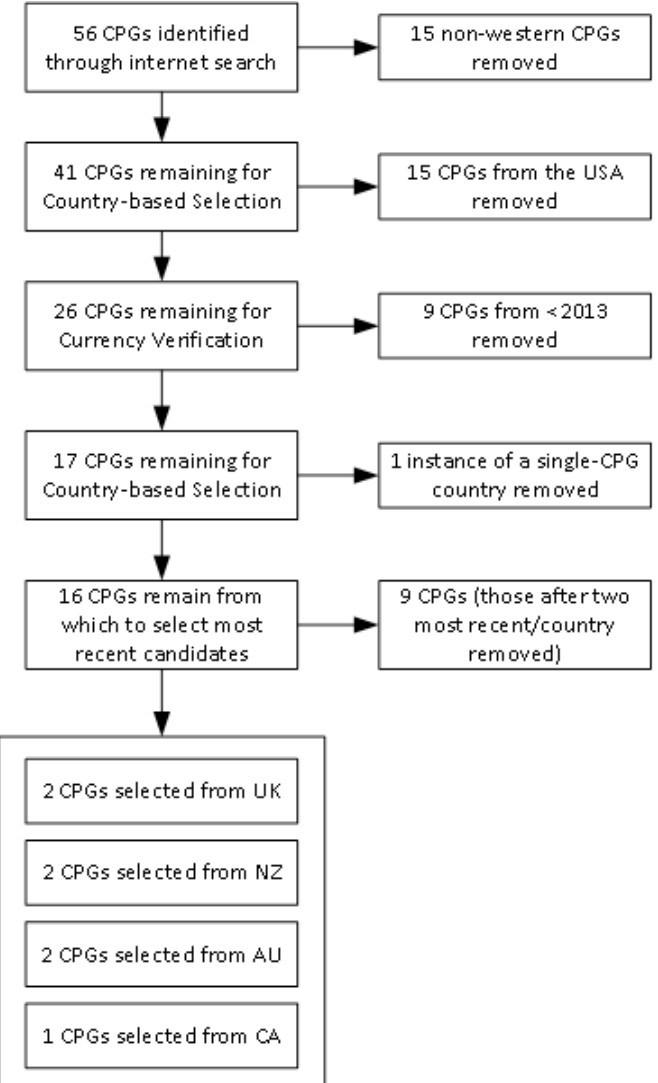

**Figure 2** CPG search and selection. CPG, clinical practice guideline.

pregnant women. The AGREE II results, once published, will be summarised on the PAMBAYESIAN website (web address) and by using social media.

## RESULTS

The internet search resulted in an initial collection of CPGs (n=56) from which we selected only those from English-speaking western tertiary hospitals—those of the UK, New Zealand (NZ), Australia (AUS), Canada (CA) and the USA—as shown in figure 2. Those from the USA were removed as, while these were sourced from hospital facilities, almost all had been produced by health insurance providers and were too limited in the identification of evidence or the evaluation process used in their production. We could not consider them independent or exclude the possibility that they were more concerned with limiting clinical test and care modalities that insurers would reimburse for rather than standardising best-practice patient care. A significant body of research identifies that good CPGs should be reviewed and updated regularly.[36] For this reason, CPGs that were not the current version, or had been produced before 2013 were also rejected. Two reviewers (BJD and SM) examined titles and abstracts to verify eligibility according to the selection criteria. Both then reviewed the full text to further verify eligibility. Given that all CA hospitals we searched (n=11) referred us to the same nationally developed CDA CPG, the authors allowed its inclusion in order to assess the reason for its clinical popularity. The resulting CPG collection included two from each of AUS, NZ and the UK, and the single CPG from CA. These are shown in table 2.

All advising bodies provide guidance on how and when screening pregnant women for GDM should ideally occur. For some, screening is performed for all women regardless of risk factors, while for others, risk stratification decides whether women are tested at booking, in early second, or during the third trimester. Risks requiring early testing include BMI >30, ethnicity, previous history of GDM, family history of diabetes and previous macrosomic baby. All CPGs included a recommendation to screen women with risk factors at booking via HbA1c to rule out pre-existing diabetes. CPGS based on the NICE guideline recommend GDM testing based on clinical risk, while those relying on CDA, BPAC, ADIPS and IADPSG

| | Country | Author organisation | Year | Title | URL |
|---|---|---|---|---|---|
| **Table 2** | Reviewed guidelines and URLs | | | | |
| 1 | AUS | Royal Women's Hospital | 2017 | Diabetes mellitus: management of gestational diabetes | https://bit.ly/2WyPXqU |
| 2 | AUS | King Edward Memorial Hospital | 2017 | Diabetes in pregnancy | https://bit.ly/2N1cAeJ |
| 3 | NZ | Auckland DHB | 2013 | Diabetes in pregnancy | https://bit.ly/2HICO6E |
| 4 | NZ | Hutt Valley DHB | 2015 | Diabetes: pre-existing and gestational | https://bit.ly/2xBuakU |
| 5 | CA | Canadian Diabetic Association | 2013 | Diabetes and pregnancy | https://bit.ly/2IdXoud |
| 6 | UK | Nottingham University Hospital | 2016 | Guideline on management of pregnant women with diabetes | http://bit.do/exsYa |
| 7 | UK | Barts Health Trust | 2015 | Diabetes - pregnancy, labour and puerperium | http://bit.do/exsXy |

recommend universal testing, or screening then testing, for all pregnant women. It should be noted that in certain locations within the UK, as a result of ethnic or cultural history, risk-based screening results in near universal testing in practice.

## CPG characteristics

Characteristics of the seven included guidelines are presented in table 3. A key issue was poor disclosure of the team involved in developing individual CPGs. Three were silent on the authoring team, and one described the authors only as the 'Diabetes Team' without further clarification. Another listed the clinical specialties of team members without identifying specific individuals. Only two (CDA and Barts Health Trust (BHT)) identified authors and their areas of clinical expertise. Only one CPG (Nottingham University Hospital (NUH)) identified that patients should be part of the treatment team. However, no guideline described consultation with service consumers or incorporated a consumer stakeholder in its development.

The scope of most CPGs in this study was limited. While all included differential diagnostic protocols and criteria, two were silent regarding antenatal steroid guidance, three provided a recommendation for consideration, and two (ADHB and BHT) provided complete guidance for managing diabetes during antenatal corticosteroid administration. In our review, two CPGs failed to provide guidance as to clinical management during labour, three provided limited guidance, and only two (KEMH and BHT) provided complete protocols for care of the diabetic mother during the labour and birth event. One CPG (HVDHB) provided no guidance for postnatal management and follow-up, while three provided limited guidance in the immediate postnatal period. The remaining three (NUH, BHT and ADHB) covered the immediate postnatal needs of the mother with diabetes, as well as providing detailed long-term follow-up.

## Appraisal of guidelines and consensus statement

We found that many CPGs did not document their individual team's method of delivering consensus, with some (from AUS RWH and UK NUH) relying on the prior consensus of an external party whose guiding protocol or evidence was used by that CPG's authors.

Table 4 reports the categorical scores for each domain calculated as per the AGREE II protocol.[32] The authors considered that CPGs with an overall score <70 require additional attention and revision to resolve issues causing that low score. For example, while both UK CPGs had three domains in which the score was 'above average', issues such as not providing in-text references to identify which evidence formed the basis for particular clinical decisions, along with limited stakeholder identification and involvement, reduced overall scores. While meeting several sections of the AGREE II protocol, the NZ HVDHB CPG also failed to identify guiding evidence for each decision, as well as neglecting to identify team members who

formulated the guideline and any potential conflicts of interest they may have had. The AUS KEMH protocol provided treatment recommendations and care plans but was exceedingly verbose and failed to identify where evidence was used or identify any single item of reference material.

Average variance provides an indication of the degree to which the reviewers gave consistent or agreeing scores for each CPG. A score close to 1 indicates little variance, which translates to strong agreement. In this study, the reviewers agreed strongly with regards to the CPGs that received the highest (CA CDA) and lowest (NZ HVDHB) AGREE II scores, as shown in table 5. The CPG with the greatest variance (AUS KEMH) resulted from one clinical reviewer scoring it higher in domains 1 and 3 (Scope and Purpose, and Rigour). That reviewer felt that, even though the evidence and references were not provided, the text demonstrated rigorous evaluation of evidence to arrive at well-devised diagnostic and treatment protocols. The remaining reviewers felt that the absence of evidence to justify decisions must reduce the overall assessment of rigour, as even the most appropriate of clinical decision requires justification through supporting evidence.

## DISCUSSION

As judged by the AGREE II protocol, all reviewed local hospital guidelines had deficiencies, notably, lack of user involvement, assessment of resource implications, listing conflicts of interests and lack of external review.

The local CPGs reviewed did not significantly diverge from the recommendations of the national or international advising bodies on which they were based. NICE and BPAC have diagnostic criteria that differ from ADIPS, CDA and IADPSG, but their residual recommendations remain consistent. The UK local CPGs broadly adhere to NICE recommendations, with refinements in one (BHT), which the CPG states is due to a population who are predominately classified as high risk. It is noted extensively in the literature that tightening the diagnostic criteria for GDM will increase the number of women diagnosed[7]; however, it remains unclear if treating the women who are outside the current NICE criteria, but within the WHO or IADPSG criteria, will result in any reduction in targeted complications sufficient to justify the increased cost to treat and increased anxiety that disclosure of potential pregnancy complications brings.[7 11 14] Given that an AGREE II protocol study concerns the form and structure of the CPG and how it was developed, such factors tend to fall outside of the protocol's remit.

While the Canadian CPG was ranked best overall in this review, it lacked in usability in application to daily care. The drafting of a practical care pathway would be required in order for midwives and obstetricians to consistently apply it in clinical practice. The other highly ranked CPG, that from NZ ADHB, provided simple but clear clinical pathway diagrams, but its use of URL links to other hospital documentation meant those sections

**Table 3** CPG characteristics

| | 1 AUS RWH | 2 AUS KEMH | 3 NZ ADHB | 4 NZ HVDHB | 5 CA CDA | 6 UK NUH | 7 UK BHT |
|---|---|---|---|---|---|---|---|
| How described by the authors | A guideline | Clinical practice guideline | Guideline | Care policy | Clinical practice guideline | Guideline | Guideline |
| Evidence and/or expert consensus based | Evidence | Evidence | Evidence | Evidence | Evidence and consensus | Evidence | Evidence |
| Clinical indication | Women with GDM | Diabetes in pregnancy | Women with diabetes in pregnancy | Pre-existing and gestational diabetes | Pre-existing and gestational diabetes | Pregnant women with diabetes (incl GDM) | Pregnant women with diabetes and GDM |
| Target users | Health professionals only (not further defined) | Not defined | All clinicians in maternity, all access holders | All midwives, obstetricians, all access holders, dieticians, endocrinologists, diabetes nurses, dietician | Not stated | All midwives, diabetic nurses. All medical staff | All Trust staff working in whatever capacity |
| Stakeholders involved | Not disclosed | Not disclosed | Diabetes team (not further defined) | Not disclosed | Broad clinical team (individual clinicians) | Broad clinical team (specialties only) | Broad clinical team (individual clinicians) |
| Interventions included | Education, self-monitoring, diet changes, prescription medication, referral to high-risk team if certain criteria are met, two weekly visits, elective delivery from 38 weeks if medicated | Education, self-monitoring, diet changes, prescription medication, referral to high-risk team if certain criteria are met, two weekly visits, elective delivery from 38 weeks if medicated, elective cesarean for macrosomia | Education, self-monitoring, diet changes, prescription medication, referral to high-risk team if certain criteria are met, 2–3 weekly visits, elective delivery from 38 weeks if medicated or uncontrolled, delivery by 41 weeks in GDM | Education, self-monitoring, prescription medication, diet changes, referral to high-risk team if certain criteria are met, 2–4 weekly visits, delivery by 41 weeks unless clinical indications for earlier delivery | Education, self-monitoring, diet changes, prescription medication | Education, self-monitoring, prescription medication, diet changes, elective delivery from 37 weeks if pre-existing, 38 weeks if uncontrolled GDM or GDM on medication, no later than 40+6 if GDM | Education, self-monitoring, prescription medication, referral to high-risk team if certain criteria are met, community pathway established, at least two weekly visits, elective delivery from 37 weeks if pre-existing, 39 weeks if GDM and medicated. Deliver by 40+6 if GDM |
| Includes in-labour management | Yes | Comprehensive | Yes | No | Yes | No | Comprehensive |
| Includes antenatal steroid management | Yes | Yes | Comprehensive | Yes | No | No | Comprehensive |
| Relied on/Referenced NICE | No | Referenced | No | Referenced | Referenced | Relied on | Relied on |

ADHB, Auckland DHB; BHT, Barts Health Trust; CDA, Canadian Diabetic Association; CPG, clinical practice guideline ; GDM, gestational diabetes mellitus; HVDHB, Hutt Valley DHB; KEMH, King Edward Memorial Hospital; NICE, The National Institute for Health and Care Excellence; NUH, Nottingham University Hospital; RWH, Royal Women's Hospital.

**Table 4** Summary of adjusted scores using the AGREE II reporting checklist

| Guideline | Scope and purpose | Stakeholder involvement | Rigour | Clarity | Applicability | Editorial independence | No. of domains above average | Overall AGREE II score |
|---|---|---|---|---|---|---|---|---|
| 1 AUS RWH | Excellent | Fair | Fair | Good | Average | Fair | 2 | 66.6 |
| 2 AUS KEMH | Average | Average | Fair | Good | Average | Average | 1 | 50 |
| 3 NZ ADHB | Excellent | Average | Average | Good | Average | Average | 2 | 83.3 |
| 4 NZ HVDHB | Excellent | Average | Average | Good | Fair | Average | 2 | 50 |
| 5 CA CDA | Excellent | Good | Good | Excellent | Good | Average | 5 | 91.6 |
| 6 UK NUH | Excellent | Good | Average | Good | Average | Fair | 3 | 66.6 |
| 7 UK BHT | Excellent | Good | Average | Good | Average | Average | 3 | 58.3 |

ADHB, Auckland DHB; AGREE II, Appraisal of Guidelines for Research and Evaluation II; BHT, Barts Health Trust; CDA, Canadian Diabetic Association; HVDHB, Hutt Valley DHB; KEMH, King Edward Memorial Hospital; NUH, Nottingham University Hospital; RWH, Royal Women's Hospital.

could only be followed by someone working within the hospital. While it provides detailed instructions for post-natal follow-up, it failed to provide appropriately detailed recommendations for fetal surveillance during the pregnancy.

At the other end of the spectrum were AUS KEMH and NZ HVDHB, which scored equal lowest in this study. One was found to be exceedingly verbose (AUS KEMH) while the other was found to be altogether too brief in its presentation (NZ HVDHB). Much of the AUS KEMH guideline concerned inpatient care, with limited support provided for care of those women in the community. The NZ HVDHB guideline provides easy-to-read careflow diagrams for the screening process but failed to provide the same for ongoing management. By contrast, those local CPGs that scored higher generally provide screening and ongoing management protocols as flow diagrams for easy review by clinicians.

The UK CPGs were produced to be generally consistent with the recommendations of the NICE guidelines. As the NICE guidelines are rigorous in their investigation and assessment of evidence, the UK guidelines scores were

**Table 5** Average variance of scores across all AGREE II domains

| Guideline | Average variance |
|---|---|
| 1 AUS RWH | 3.07 |
| 2 AUS KEMH | 3.69 |
| 3 NZ ADHB | 2.03 |
| 4 NZ HVDHB | 1.65 |
| 5 CA CDA | 1.08 |
| 6 UK NUH | 2.09 |
| 7 UK BHT | 2.23 |

ADHB, Auckland DHB; AGREE II, Appraisal of Guidelines for Research and Evaluation II; BHT, Barts Health Trust; CDA, Canadian Diabetic Association; HVDHB, Hutt Valley DHB; KEMH, King Edward Memorial Hospital; NUH, Nottingham University Hospital; RWH, Royal Women's Hospital.

not impacted so much by the lack of evidence, but by the style and lack of adherence to the format prescribed by the AGREE II protocol. We note that such organisations often produce CPGs based on an internal template or style guide, and as such, issues in this area should be addressed as a matter for institutional change.

A key finding of our study was that consistent deficiencies and inconsistencies permeate guidelines for GDM care; a finding that is only possible when reviewing a representative sample, and a key motivator for why we selected guidelines across a number of developed western countries.

All local CPGs in this study provided glycaemic control targets at levels equal to or better than their respective diagnostic criteria for GDM. Similarly, all CPGs included guidance on initiation of medication (metformin and/or insulin) if diet modification and exercise failed to achieve glycaemic control. One CPG (AUS KEMH) provided extensive guidance on education and administration of insulin for the patient. All CPGs recommended the transfer of GDM patients requiring medication into a high-risk or multidisciplinary care team, with provision for diet-controlled women to remain in their community teams with secondary care oversight. All CPGs encouraged community, primary care models for diet-controlled GDM and allowed the pregnancy to continue to 41 weeks' gestation before recommending elective delivery. This reduces the number of interventions for women with well-controlled GDM. All CPGs agreed on earlier elective delivery (from 38 to 39 weeks' gestation), if the woman required medication to achieve glycaemic control.

This study found that fetal surveillance recommendations varied markedly: CDA contained no advice on ultrasound or fetal condition monitoring during pregnancy; ADHB stated that GDM in itself was not a reason for increased fetal surveillance and accordingly made no specific recommendations; most other CPGs recommended some form of ultrasound monitoring if the woman required medication for glycaemic control. It should be noted that none of the seven advising bodies

listed in table 1 included recommendations for fetal surveillance in GDM pregnancies in their consensus statements.

In summary, this review analysed seven current hospital CPGs addressing the diagnosis and management of GDM and confirmed significant variation on the quality of the local guidelines. Only two CPGs (CA CDA and NZ ADHB) were considered by the authors to be of acceptable quality when assessed using the AGREE II criteria, demonstrated by achieving a score >70. Two received a score of 50 (AUS KEMH and NZ HVDHB) and were regarded as flawed by the authors. For the remaining three (AUS RWH, UK NUH and UK BHT), minimal work would be required to improve the template and presentation approach in order to meet AGREE II standards, as no significant issue could be raised with their clinical recommendations which adhered to suitable advising body guidelines.

There are a number of limitations to this study. Only two selected CPGs were selected from tertiary hospitals and from only three English-speaking countries; furthermore, CPGs before 2013 were not considered, even though in many settings they are still used in current clinical practice. Nonetheless, the range of AGREE II scores suggests that we captured the range of available guidelines thus confirming that there is wide variation in local guidelines. While the AGREE II protocol sets a minimum of two reviewers, but preferably more for a successful protocol review, our study used five. Given the range of qualifications and experience of the reviewers in this study, we believe this significantly adds to the robustness of the results. We restricted our evaluation to GDM and did not include the management of pre-existing diabetes; this narrowed down the available guidelines for review. An alternate approach would have been to evaluate all the international guidelines. However, in a tertiary setting, they are interpreted locally leading to the guidelines used in the local clinical setting, which was the subject of our review. Finally, the AGREE II protocol does not assess the evidence base of the clinical content of the guideline, or its implementation. Thus, a guideline reviewed may score highly, independent of the local or national adaptation of the international guideline but does not indicate the clinical quality of the decision making or the evidence or how it was adapted for local needs.

## CONCLUSION

Good quality local CPGs provide complete care for the identified patient cohort during the identified health incident (in this case, pregnant women with GDM). Authors of CPGs should ensure that the evidence relied on to guide clinical decisions within the CPG is directly referenced so that users can be assured of the rigour and appropriateness of the recommendation. This was a key area lacking in almost all of the CPGs reviewed. CPGs should not omit steps relevant to the care of their patient cohort. Nor should they require additional time and effort on the part of clinicians to seek out sections of other CPGs or clinical artefacts (such as a separate labour and birth guideline with a section dealing with delivery of the diabetic mother). Although each of the CPGs reviewed in this study was found to be a complete guideline, the degree of detail and justification provided in some requires attention.

Future development of local CPGs should include a clear listing of those who have undertaken development of the guideline, user involvement, an assessment of resource implications, a listing of conflicts of interests and external review.

**Acknowledgements** The North East London Diabetes Research Network Lay Panel are thanked for their contribution to this project.

**Contributors** BD and SM proposed guidelines for inclusion. SM prepared survey to AGREE II requirements. All authors reviewed selected guidelines. BD and SM composed initial draft. GH and BD prepared and revised clinical content. SM prepared results spreadsheet. NF reviewed results and performed AGREE II and variance calculations. All authors commented and approved the final draft.

**Funding** All authors acknowledge support from the Engineering and Physical Sciences Research Council (EPSRC) under project EP/P009964/1: PAMBAYESIAN: Patient Managed decision-support using Bayes Networks.

**Competing interests** None Declared.

**Patient consent for publication** Not required.

**Provenance and peer review** Not commissioned; externally peer reviewed.

**Data sharing statement** Data available on request.

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
