## [Reviewer comments · BMJ Open]

ARTICLE DETAILS

TITLE (PROVISIONAL)	Assessment of the Methodological Quality of Local Clinical Practice Guidelines on the Identification and Management of Gestational Diabetes
AUTHORS	Daley, Bridget; Hitman, Graham; Fenton, Norman; McLachlan, Scott

VERSION 1 - REVIEW

REVIEWER	Basilio Pintaudi Diabetes Unit, Niguarda Hospital, Milan, Italy
REVIEW RETURNED	16-Nov-2018

GENERAL COMMENTS	Daley B and Colleagues performed a systematic review of clinical practice guidelines (CPGs) for diabetes in pregnancy using the Appraisal of Guidelines for Research and Education (AGREE II) to assess and validate methodological quality. Among the 7 identified CPGs Authors found that only one CPG scored above average in five or more of the six AGREE II domains. Significant variation existed in the methodological quality of CPGs. Appropriate identification of the evidence relied on to inform clinical decision making in CPGs was poor, as was evidence of user involvement in the development of the guideline, resource implications, documentation of competing interests of the guideline development group and evidence of external review. Overall, the paper is well-done and it could give additional elements to the international literature on clinical practice guidelines on the identification and management of diabetes in pregnancy. However, the Authors must solve the following points: 1) no more than two CPGs were selected from any single country. How were they selected in the case of the presence of more than two?2) the introduction section of the manuscript is too long. It should be shortened.3) one of the main limit of the review is having selected CPGs only from tertiary hospitals and from some Countries (UK, New Zealand, Australia, Canada, and the United States of America). CPGs before 2013 were also rejected. These aspects should be recognized as study limitations.4) at the end of pag 11 there is written: "it should be noted that none of the six advising bodies listed in Table 4...". In table 4 there are 7 guidelines. Please check.
--

	5) Authors should acknowledge that some CGPs on diabetes includes recommendations for diabetes in pregnancy. These are not included in the review, this limiting the number of considered CGPs.
--	---

REVIEWER	Mukesh M Agarwal California University of Science and Medicine USA
REVIEW RETURNED	18-Dec-2018

GENERAL COMMENTS	Daley et al. take on the valiant task of evaluating guidelines for gestational diabetes (GDM) applying an objective tool for guidelines: AGREE II. General comments: Currently, there are a spate of guidelines for GDM from major pre-eminent medical associations: national (e.g., Canadian Diabetes Association, CDA; Brazil; France); international (World Health Organization, WHO; Federation of International Gynecology and Obstetrics, FIGO); obstetric organizations (American College of Gynecologists, ACOG) and diabetes expert panels (American Diabetes Organization, ADA). Also, most (but not all) guidelines have evolved over the years (e.g., WHO-1985, WHO-1999, WHO-2013) incorporating the latest research. The result is simple: chaos. As expected, these multiple approaches to GDM confuse the health provider, who is looking for clarity in approaching GDM. Thus, in the same hospital, despite recommended guidelines by the hospital, multiple different guidelines could be in use—and sometimes even older, obsolete guidelines, which beats all logic. Major comments:  1. The main problem with this paper is that most of the guidelines that are evaluated are too parochial and empirical to be of use to any caregiver of pregnant women—in any hospital in the world. Currently, the major valid GDM guidelines are of a) The International Association of the Diabetes and Pregnancy Study Groups, IADPSG-2010 (accepted by WHO, ADA, ADIPS, FIGO amongst many others—and the most popular), b) The National Institute for Health and Care Excellence, NICE-2015, c) Canadian diabetes association, CDA-2013. In North America, Carpenter & Coustan, C&C and National Diabetes Data Group, NGGD-1979 (accepted by ACOG/ADA) are very popular. 2. The authors recommend two guidelines of acceptable quality based on their assessment tool: CDA and New Zealand guidelines, NZ. Both these guidelines are insular and ‘expert-opinion driven’ without any research backing. The CDA raised the HAPO study threshold 1.75 risk to 2.0 odds-risk ratio without universal consensus (unlike IADPSG). Furthermore, CDA-2013 guidelines are a hybrid using the 50-g screening and 75-g OGTT which is validated by no major study. The NZ guidelines are worse for 2 reasons: a) they were based on the now defunct ADIPS criteria and b) their 2-hr 75-g OGTT value was raised from 8.0 mmol/l of ADIPS to 9.0 mmol/l to save ‘New Zealand’s stretched funds.’ In short it was an economic decision. Furthermore, the NZ guidelines have not evolved using the latest research.
--

	3. The NICE-2015 guidelines used by the authors were based on 'economics' rather than 'research.' Thus, many major hospitals, despite the clout of NICE, shy away from their GDM guidelines. 4. Due to the authors' restrictions in criteria selection (reject CPGs before 2013), the authors eliminated one of the most popular guidelines vogue in USA (C&C) – which has stood the test of time. Minor comments:  1. The authors make a good point that weakest link in the chain of practice guidelines is implementing the guideline. Thus, the recommendation of the latest FIGO guidelines to be 'practical.' 2. The author's reference 1 has been retracted. Please replace by an alternate one. 3. The references should be cut to less than 20. In short, none of the guidelines suggested by the authors are not validated by any research. Their suggested guidelines may be meet all the criteria of AGREE II, but their conclusions would be a hard sell to any hospital. Somehow, this paper would have great value if the authors had evaluated the current (WHO-2013, C&C, ACOG, NICE-2015, CDA-2013) vogue and more acceptable guidelines using their instrument.
--	---

REVIEWER	Jiemin Pan Dept. of Endocrinology & Metabolism, Shanghai Sixth People's Hospital affiliated to Shanghai Jiaotong University, Shanghai, China
REVIEW RETURNED	20-Dec-2018

GENERAL COMMENTS	This is a systematic review of clinical guidelines for diabetes in pregnancy using the Appraisal of Guidelines for Research and Evaluation (AGREE II) instrument to assess and validate their methodological quality. Of 56 CPGs identified, 7 CPGs were evaluated in detail by five reviewers. Results presented by a five-point Likert scale, the authors noted that appropriate identification of the evidence relied on to inform clinical decision making in CPGs was poor, as was evidence of user involvement in the development of the guideline, resource implications, documentation of competing interests of the guideline development group and evidence of external review. Major Comments: Abstract :  1. Part of the content in Results is more like subjective opinion, thus should be removed to Conclusion, or try a more objective way. 2. Data Sources. Do you think that your criteria of inclusion can guarantee representative or diversity? 3. Data Sources. Page2Line18: "(5) demonstrated evaluation or inclusion of evidence", can you explain the meaning of it? 4. What do you think "Evaluation of each guideline by five reviewers"? Is it the strength or limitation? Introduction:  1. It's kind of wordy and unclear. It would be better if the authors can re-organize it. 2. Page4Line26: There are missing spaces at the beginning of the paragraph.
--

3. The format of table 1, as well as table 2-4 afterwards, should be a three-line table.

Methods:

1. Guideline Selection. Apart from selection criteria mentioned, maybe you can add your search strategy including database and mesh terms to improve the representative of your guideline. And accordingly add a flowchart in the Result.

2. The Review. Maybe the subtitle can be changed to "Guideline quality assessment".

3. Here you should explain how these data are processed, that is, the statistical analysis method.

Results:

1. As a general rule, CPGs should be reassessed for validity every 3 years[1, 2].

Page 6Line26: Here you mentioned "CPGs that were not the current version, or had been produced before 2013 were also rejected.", what's the basis of it?

2. Page7Line5: There are missing spaces at the beginning of the paragraph.

Discussion:

1. The guidelines talk about diabetes in pregnancy, however, the authors uses a lot "GDM" instead especially in "Clinical Management", which is inappropriate. The authors may check it and find another more suitable way.

2. The authors may add something about AGREE II in the Discussion, like strengths and limitations, to enhance the integrity of the article.

Conclusion:

1. Page12: the first paragraph is more like part of results rather than conclusion. The authors may reconsider it and better organize it.

2. In general, it's kind of wordy and would be better if it's more concise.

Minor Comments :

The manuscript needs to be edited for grammar and syntax.

For instance:

(1) Page2Line4: "clinical practice guidelines (CPG)" should be rewritten as "clinical practice guidelines (CPGs)".

(2) Page2Line10: "the Appraisal of Guidelines for Research and Education (AGREE II)" should be rewritten as "the Appraisal of Guidelines for Research and Evaluation (AGREE II)".

(3) Page2Line48: "non-english" should be rewritten as "non-English".

(4) Page3Line17: "6-to-11-fold" should be rewritten as "6- to 11-fold".

(5) Page4Line37: "the expense of repeat or inappropriate treatments" should be rewritten as "the expense of repeating inappropriate treatments".

(6) Page5Line4: "23 questions" would be better if written as "23 items".

(7) Page5Line52: "clinical midwife research fellow" should be rewritten as "a clinical midwife research fellow".

(8) Page7Line26: "treating staff" should be expressed in another word, like "clinician", for it isn't a customary usage.

(9) Page11Line15: "treating clinicians" should be expressed in another way, reason mentioned above.

(10) Page2Line29: "Prior to the introduction of insulin in 1923 maternal and neonatal ...", there ought to be a comma after "in 1923", that is, "Prior to the introduction of insulin in 1923, maternal and neonatal ...",

	(11) Page5Line49: "To increase the reliability of the appraisal the AGREE II protocol user manual...", there ought to be a comma after "the appraisal", that is, "To increase the reliability of the appraisal, the AGREE II protocol user manual...". Ref.:  1. Shekelle, P., et al., When should clinical guidelines be updated? Bmj, 2001. 323(7305): p. 155-7. 2. Shekelle, P.G., et al., Validity of the Agency for Healthcare Research and Quality clinical practice guidelines: how quickly do guidelines become outdated? Jama, 2001. 286(12): p. 1461-7.
--	---

VERSION 1 – AUTHOR RESPONSE

Reviewer 1:

1. Overall, the paper is well-done

Thank you

2. No more than two CPGs were selected from any single country. How were they selected in the case of the presence of more than two?

The selection process is described in the paper under the heading Guideline Selection. We have now addressed the comment regarding selection of 2 CPGs per country where more than 2 met the criteria at the end of that section. In the methods we have stated (page 5 line 26)

"Where more than two from any one country met the requirements, the ones having the most recent updates were used."

We have also added a new section in the discussion (page 11 line 18) on limitations of the study

"There are a number of study limitations to this study. Only two selected CPGs were selected from tertiary hospitals and from only 4 English-speaking countries; furthermore, CPGs before 2013 not considered, even though in many settings they are still used in current clinical practice. Nonetheless, the range of AGREE II scores would suggest we captured the range of available guidelines thus confirming that there is wide variation in local guidelines."

3. The introduction section of the manuscript is too long. It should be shortened.

Content has been revised and shortened. Overall the paper has been reduced from 6839 to 6593 words, a total reduction of 246 with 237 of that in the content of the introduction.

4. One of the main limits of the review is having selected CPGs only from tertiary hospitals and from some Countries (UK, New Zealand, Australia, Canada, and the United States of America). CPGs before 2013 were also rejected. These aspects should be recognized as study limitations.

We have added a notation to the Strengths and Limitations section.'

"Exclusion of CPGs prior to 2013 and limiting to the most recent two located in each country means that while those CPGs being reviewed are the most up-to-date, some well-known older guidelines were excluded."

We have also discussed this under study limitations in the discussion; see comments reviewer 1 (2).

5. At the end of page 11 there is written: "it should be noted that none of the six advising bodies listed in Table 4...". In table 4 there are 7 guidelines. Please check.

Addressed. Corrected to:

"seven advising bodies in Table 1"

6. Authors should acknowledge that some CGPs on diabetes includes recommendations for diabetes in pregnancy. These are not included in the review, this limiting the number of considered CGPs

Addressed. The authors are only addressing the diagnosis and management of gestational diabetes (GDM) not "diabetes in pregnancy" which includes pre-existing T1 and T2 diabetes.

This has been made clearer in the Abstract, Introduction and Methods sections. In the discussion we have added the sentence referring to limitations of the study (page 11line 22)

"We restricted our evaluation to GDM and did not include the management of pre-existing diabetes; this narrowed down the available guidelines for review."

Reviewer 2:

1. The main problem with this paper is that most of the guidelines that are evaluated are too parochial and empirical to be of use to any caregiver of pregnant women- in any hospital in the world.

The reviewer has a valid point and indeed we are not recommending the adoption of any one of the guidelines reviewed but pointing out the limitations of guideline development and the areas that can be improved using the AGREE II protocol. As stated in the paper the authors reviewed local

guidelines that are IN USE in hospitals in countries in which the authors have clinical experience, and which operate a centralised health service. Rather than review the overarching international guidelines (ADIPS, IADPSG, NICE etc.) which have already been extensively reviewed and analysed in the common literature, we sought to evaluate the guidelines as they are being prescribed locally within tertiary hospitals. We have also added under study limitations in the discussion the following sentences (page 11 line 24)

“An alternate approach would have to evaluate all the international guidelines, however, in a tertiary setting they are interpreted locally leading to the guidelines used in the clinical setting, that was the subject of our review. Finally, the AGREE II protocol does not assess the evidence base of the clinical content of the guideline, nor its implementation. Thus, a guideline reviewed may score highly, independent of the local or national adaptation of the international guideline but does not indicate the clinical quality of the decision making or the evidence or how it was adapted for local needs.”

2. The authors recommend two guidelines of acceptable quality based on their assessment tool: CDA and New Zealand guidelines, NZ. Both these guidelines are insular and ‘expert-opinion driven’ without any research backing.

Our response is covered above in answer to Reviewer 2 Q1 – the purpose of the AGREE II protocol is not to identify the guideline with the best content but to analyse the process used to formulate the guideline using the AGREE II tool. Furthermore Domain 3 of the AGREE II protocol, regarding the rigour and evidence for a CPG, is only one of the areas that AGREE II evaluates. CDA-2013 scored well across all Domains in the opinions of the reviewers. NZ-ADHB scored “average” in this Domain, but maintained a better score across the other Domains than the other CPGs reviewed. NZ-ADHB also presented the evidence that they relied upon and explained decisions taken in formulating their recommendations, allowing readers and clinicians to evaluate for themselves the recommendations provided.

3. The NICE-2015 guidelines used by the authors were based on ‘economics’ rather than ‘research.’ Thus, many major hospitals, despite the clout of NICE, shy away from their GDM guidelines.

Again, this is a valid point and indeed all international guidelines have their drawbacks leading to the confusion in the field as to which should be adopted. Unfortunately for the user of the service, affordability does play a role in the development of the guideline and its parochial interpretation.

4. Due to the authors’ restrictions in criteria selection (reject CPGs before 2013), the authors eliminated one of the most popular guidelines vogue in USA (C&C) – which has stood the test of time.

We respect the reviewer’s opinion. In the limitations section of the discussion we have therefore added...

“CPGs before 2013 not considered”

...even though in many settings they are still used in current clinical practice. Nonetheless, the range of AGREE II scores would suggest we captured the range of available guidelines thus confirming that there is wide variation in local guidelines.”

5. The authors make a good point that weakest link in the chain of practice guidelines is implementing the guideline. Thus, the recommendation of the latest FIGO guidelines to be ‘practical.’

No response

6. The author’s reference 1 has been retracted. Please replace by an alternate one.

The article itself remains available on the publisher’s database and web archive. In any event the article was not retracted for any factual error or omission of the authors, it was retracted simply because, in the assessment of an editor, the content extensively overlapped with a number of prior publications.

The reference has been updated and the new reference 1 is :

Metzger, B. and D. Coustan. (1998). Proceedings of the Fourth International Work-shop-Conference on Gestational Diabetes Mellitus. *Diabetes Care*, 21, B1-B167.

7. The references should be cut to less than 20.

We have reduced the references from 56 to 36

8. In short, none of the guidelines suggested by the authors are not validated by any research.

We agree with the reviewer that rarely are locally adapted guidelines entirely evidence based and have been adapted by local considerations. The AGREE II study evaluates this under Domain 3 of the AGREE II protocol that specifically addresses this issue as part of assessing the rigour of development. All the CPGs reviewed identified which international diagnostic and treatment guidelines they evaluated for inclusion (eg IADPSG, NICE, ADIPS).

Reviewer 3:

Major Comments:

Abstract :

1. Part of the content in Results is more like subjective opinion thus should be removed to Conclusion or try a more objective way.

We have edited the results section and removed the subjective or interpreted statements we believe the reviewer eludes to. Examples of deleted text are below

“Not knowing who was responsible for developing a guideline makes it difficult to know who is the most appropriate person that clinicians can approach when they have questions about guideline content. It also makes post hoc identification and assessment of conflicts of interest and their impact on the guideline’s development more difficult.”

“It is inefficient to have guidelines that require clinicians to refer to other guidelines when known points in the pregnancy are reached, especially when those guidelines have also had to be modified to incorporate sections for care specific to the first guideline’s patients or condition.”

2. Data Sources. Do you think that your criteria of inclusion can guarantee representative or diversity?

Yes. We sampled different cities in different countries with different representative populations.

3. Data Sources. Page2Line18: “(5) demonstrated evaluation or inclusion of evidence”, can you explain the meaning of it?

Whilst we are comfortable with this statement we have replaced the text with

‘Grounded on evidence-based medicine’ which we hope will be clearer to the reader

.

4. What do you think “Evaluation of each guideline by five reviewers”? Is it the strength or limitation?

This is a very difficult question as we are unaware of power calculations that would enable the answer. We have therefore included the following sentences under limitations in the discussion

‘While the AGREE II protocol sets a minimum of two reviewers, but preferably more, for a successful protocol review, our study used five. Given the range of qualifications and experience of the reviewers in this study, we believe this significantly adds to the robustness of the results. ‘

Introduction:

1. It's kind of wordy and unclear. It would be better if the authors can re-organize it.

The Introduction has been restructured. See comment from Reviewer 1 above.

2. Page4Line26: There are missing spaces at the beginning of the paragraph.

Addressed. Spaces added.

3. The format of table 1, as well as table 2-4 afterwards, should be a three-line table.

Methods:

1. Guideline Selection. Apart from selection criteria mentioned, maybe you can add your search strategy including database and mesh terms to improve the representative of your guideline. And accordingly add a flowchart in the Result.

This was a search for publicly accessible guidelines and our preference as stated in the paper was for those in clinical practice at hospitals. Hospital websites were the primary source, and these are almost never indexed by PubMed or MeSH.

2. The Review. Maybe the subtitle can be changed to "Guideline quality assessment".

We have changed the heading as suggested.

3. Here you should explain how these data are processed, that is, the statistical analysis method.

The section labelled "The Review" addresses the methods by which the results were analyzed but at the reviewer's suggestion have further elucidated on the statistics as below (page 5 line 34)

'Responses were collected using a secure online survey tool that exported into an Excel spreadsheet prepared specifically to report per domain, per question and per reviewer. Percentages were calculated in Excel for each domain following the AGREE II protocol manual algorithm shown in

Figure 1. Additionally, reviewer variance scores were calculated across all scores for each question using Excel's integrated VAR function.'

Figure 1: AGREE II Protocol Domain Scoring Algorithm

Results:

1. As a general rule, CPGs should be reassessed for validity every 3 years[1, 2].

Page 6Line26: Here you mentioned "CPGs that were not the current version, or had been produced before 2013 were also rejected.", what's the basis of it?

Addressed. See response to Reviewer 2 Q4.

2. Page7Line5: There are missing spaces at the beginning of the paragraph.

Addressed. Spaces added where indicated

Discussion:

1. The guidelines talk about diabetes in pregnancy, however, the authors use a lot "GDM" instead especially in "Clinical Management", which is inappropriate. The authors may check it and find another more suitable way.

Addressed. The authors are only addressing the diagnosis and management of gestational diabetes (GDM) not "diabetes in pregnancy" which includes pre-existing T1 and T2 diabetes. See response to Reviewer 1 above.

2. The authors may add something about AGREE II in the Discussion like strengths and limitations, to enhance the integrity of the article.

This was addressed in the section "Clinical Practice Guidelines", para 3 page 4 and now in the discussion under limitations of the study. We have also added a new paragraph in the discussion as below (page 11 line 18):

"There are a number of study limitations to this study. Only two selected CPGs were selected from tertiary hospitals and from only 3 English-speaking countries; furthermore, CPGs before 2013 not considered, even though in many settings they are still used in current clinical practice. Nonetheless, the range of AGREE II scores would suggest we captured the range of available guidelines thus confirming that there is wide variation in local guidelines. While the AGREE II protocol sets a

minimum of two reviewers, but preferably more, for a successful protocol review, our study used five. Given the range of qualifications and experience of the reviewers in this study, we believe this significantly adds to the robustness of the results. We restricted our evaluation to GDM and did not include the management of pre-existing diabetes; this narrowed down the available guidelines for review. An alternate approach would have to evaluate all the international guidelines, however, in a tertiary setting they are interpreted locally leading to the guidelines used in the local clinical setting, that was the subject of our review. Finally, the AGREE II protocol does not assess the evidence base of the clinical content of the guideline, nor its implementation. Thus, a guideline reviewed may score highly, independent of the local or national adaptation of the international guideline but does not indicate the clinical quality of the decision making or the evidence or how it was adapted for local needs.”

Conclusion:

1. Page12: the first paragraph is more like part of results rather than conclusion. The authors may reconsider it and better organize it.

This paragraph was moved to the end of the Discussion as it summarises the findings that were discussed.

3. In general, it's kind of wordy and would be better if it's more concise.

We have edited the paper accordingly and in doing so reduced the word count and made the individual sections more concise.

Minor Comments :

The manuscript needs to be edited for grammar and syntax.

For instance:

1. Page2Line4: “clinical practice guidelines (CPG)” should be rewritten as “clinical practice guidelines (CPGs)”.

Addressed. Changed as per advice.

2. Page2Line10: “the Appraisal of Guidelines for Research and Education (AGREE II)” should be rewritten as “the Appraisal of Guidelines for Research and Evaluation (AGREE II)”.

Addressed. Changed as per advice

3. Page2Line48: "non-english" should be rewritten as "non-English".

Addressed. Changed as per advice.

4. Page3Line17: "6-to-11-fold" should be rewritten as "6- to 11-fold".

Addressed. Changed as per advice.

5. Page4Line37: "the expense of repeat or inappropriate treatments" should be rewritten as "the expense of repeating inappropriate treatments".

Addressed. Changed as per advice.

6. Page5Line4: "23 questions" would be better if written as "23 items".

Addressed. Changed as per advice.

7. Page5Line52: "clinical midwife research fellow" should be rewritten as "a clinical midwife research fellow".

Addressed. Changed as per advice.

8. Page7Line26: "treating staff" should be expressed in another word, like "clinician", for it isn't a customary usage.

Addressed. Changed as per advice.

9. Page11Line15: "treating clinicians" should be expressed in another way, reason mentioned above.

Addressed. Changed as per advice.

10. Page2Line29: "Prior to the introduction of insulin in 1923 maternal and neonatal ...", there ought to be a comma after "in 1923", that is, "Prior to the introduction of insulin in 1923, maternal and neonatal...",

Addressed. Changed as per advice.

11. Page5Line49: "To increase the reliability of the appraisal the AGREE II protocol user manual...", there ought to be a comma after "the appraisal", that is, "To increase the reliability of the appraisal, the AGREEII protocol user manual...".

Addressed. Changed as per advice.

VERSION 2 – REVIEW

REVIEWER	Jiemin Pan Department of Endocrinology & Metabolism, Shanghai Sixth People's Hospital, Shanghai, China
REVIEW RETURNED	11-Mar-2019

GENERAL COMMENTS	well modified, but many grammatical errors need to be corrected
---

VERSION 2 – AUTHOR RESPONSE

Reviewer 3: well modified, but many grammatical errors need to be corrected

We have proof read the manuscript and made relevant changes